# Analysis of Load Transfer and the Law of Deformation within a Pile-Supported Reinforced Embankment

**Da Zhang** [1,2]**, Guangqing Yang** [1,2,3,]*****, Xin Wang** [1,4,] **Zhijie Wang** [1,2,3] **and He Wang** [1,2,3]

1 State Key Laboratory of Mechanical Behavior and System Safety of Traffic Engineering Structures, Shijiazhuang Tiedao University, Shijiazhuang 050043, China
2 School of Civil Engineering, Shijiazhuang Tiedao University, Shijiazhuang 050043, China
3 Hebei Engineering Research Center on Application of Geosynthetics, Shijiazhuang 050043, China
4 School of Traffic and Transportation, Shijiazhuang Tiedao University, Shijiazhuang 050043, China
* Correspondence: yanggq@stdu.edu.cn; Tel.: +86-139-3112-0930

**Abstract:** In this paper, the load-transfer mechanism and settlement behaviors of the pile-supported reinforced embankment are reviewed by laboratory model tests, and a series of finite element method (FEM) modellings are conducted to analyze the soil-arching geometry and embankment deformation patterns of the pile-supported reinforced embankment. The results show that: the embankment load distribution is significantly impacted by the filling cohesion because of the effect of cohesion on the interaction between particles. The soil pressure difference between the center and corner of the pile caps decreases with the increase of filling cohesion. The pile-subsoil stress ratio decreases with the increase of filling cohesion. The embankment deformation behavior and soil-arching geometry are less affected by the change in filling cohesion compared with the influence of pile spacing. That may because of the fact that although the cohesion of the embankment filling has been increased, the granular material's properties have not been fundamentally changed. Pile-subsoil different settlement decreases with the increase of embankment filling cohesion, and the different settlement at the mid-span between four piles decreases by 4.09% and 6.34%, respectively, as filling cohesion increases from 0 kPa to 11 kPa and 25 kPa. The height of the soil-arching crown decreases with the increase of filling cohesion, and the height of the soil-arching crown between horizontal adjacent piles decreases by 3.85%, 7.69%, and 9.62%, as filling cohesion increases from 5 kPa to 15 kPa, 25 kPa and 45 kPa. The rate of decrease in soil-arching height gradually decreases with increasing cohesion. The height of the soil-arching between the horizontal adjacent piles is about $1.0\,(s-a)$. The height of soil arching between the diagonal adjacent piles is about $1.0\sqrt{2}\,(s-a)$. The differential settlement at the same height inside the embankment decreases with the increase of filling cohesion, and the height of the equal settlement plane is basically the same as the height of soil arching.

**Keywords:** pile-supported reinforced embankment; model test; soil arching effect; cohesion; equal settlement plane

## 1. Introduction

The design solution of pile-supported reinforced embankment has been frequently used to overcome issues (stability and settlement) encountered in soft subsoil areas [1]. The embankment load is mainly shared by the piles and a bearing stratum through the soil-arching effect and the membrane effect. Therefore, it is possible to effectively control the post-construction settlement. On the basis of theoretical analysis, field testing, model tests, and simulation studies, a series of studies on the load-transfer mechanism and the deformation patterns of soil arching have been carried out by numerous academics. Due to the intricacy of its functional properties, there are still gaps in our understanding of the soil-arching effect.

　　　　Numerous approaches have been offered to calculate the load transfer efficiency of the soil-arching effect. Marston [2] established the theoretical model of soil arching, which is based on the redistribution phenomenon of the stress above rigid pipes. Liu Jifu [3] established a model in which the soil is divided into inner cylindrical and outer cylindrical soil, and deduced a mathematical model for calculating load transfer efficiency based on Marston theory. Zhang Chengfu et al. [4] improved Liu's approach by considering the friction coefficient β between the inner and outer soil columns is non-constant. Terzaghi [5] revealed that soil arching is, in essence, a phenomenon of stress distribution, and proposed a two-dimensional soil arching model through various trapdoor tests. It was later extended by Russell and Pierpoint [6] to a three-dimensional soil arching model. Hewlett and Randolph [7] put forward a semi-spherical model in response to model tests, assuming that the crown or feet of arching are in the ultimate state. The British standard, BS8006-1 [8] adopted both this model and Marston's theory. Zeaske & Kempfert [9] improved the H&R model and proposed a multi-arching model composed of semi-spheres with different centers, which was incorporated into the German standard, EBGEO [10]. Furthermore, Eekelen [11] found that the load distribution above the reinforcement is an inverted triangle through the test and proposed an arching model with concentric hemispheres. Carlson [12] argued that the foundation soil between piles only bore the weight of the wedged soil and established a wedged soil arching model under this assumption, and the model was adopted by the Nordic Design Guide [13]. Related research [14] shows that the existing soil arching models used to calculate the pile load-sharing ratio differ from each other because of the different assumptions.

　　　　The physical model test, a reliable way to evaluate the operational performance of pile-supported reinforced embankments, has been widely used. Numerous laboratory model tests based on the trapdoor test have been conducted by several academics. Iglesia et al. [15] used coarse sand to fill the centrifuge tests. They investigated into how soil arching changed as a result of internal friction angle of filling and embankment height. Dry sand was utilized to fill indoor model tests by Cao Weiping [16] and Fei Kang et al. [17] in order to study the impact of pile spacing and embankment height on the embankment's load-transfer mechanism. Essar [18] studied arching with different fillers (Toyoura sand, silica sand, and dry powder clay) and used X-ray CT scanning. Fang [19] and Wang [20] used the photoelastic experiment technology, which took the photoelastic material as the filling, and studied the evolution process of macroscopic soil arching effect morphology. Zhang Zhen et al. [21] used transparent soil as filler, mainly composed of high-purity quartz particles, and studied the evolution of arching under static load. Rui et al. [22,23] respectively took elliptical steel bars and medium-coarse grade quartz sand as the embankment filling, investigated the deformation model of embankment with and without reinforcement, and explored the load distribution mechanism of the embankment under different tensile strengths of reinforcement.

　　　　The embankment fillings used in the above model tests are mainly sand or similar, non-cohesive soil. Few studies have examined how filling cohesiveness affects the pile-supported reinforced embankment's deformation mode, load transmission mechanism, and soil arching form. However, most embankment filling in realistic engineering have cohesion. More importantly, it is also one of the non-negligible components of shear stress. Consequently, investigating how filling cohesion affects the load-transfer mechanism and deformation model is essential. The load transmission mechanism, settling behavior, and evolution law of soil arching form under various cohesions of embankment filling are studied in this research using model tests and the finite element method (FEM). The specific research process can be seen in Figure 1.

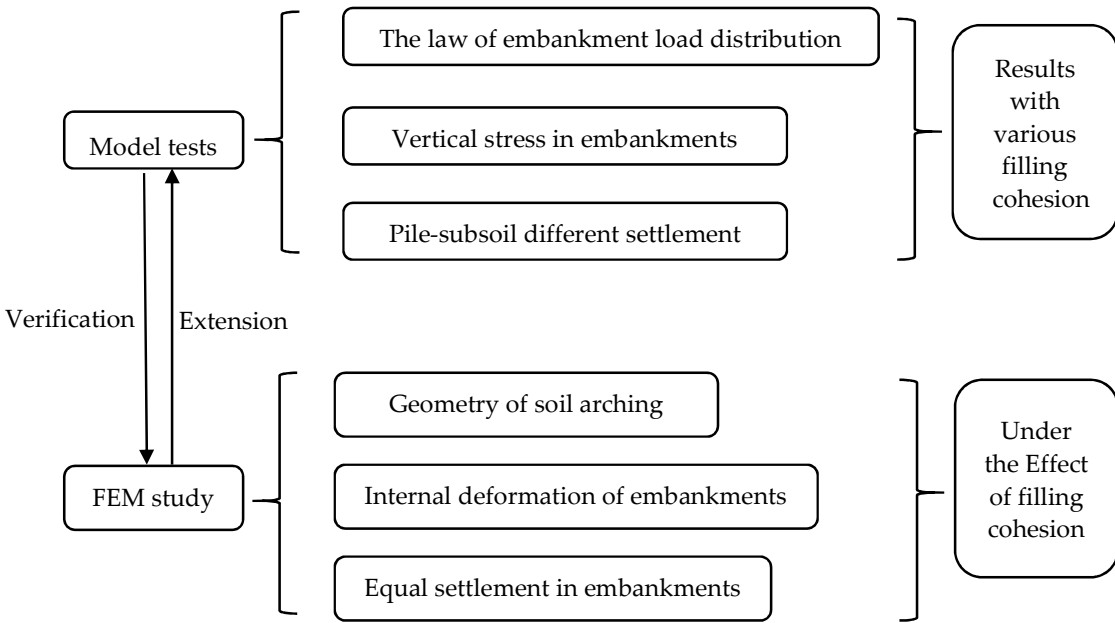

**Figure 1.** The flowchart of the research methodology.

## 2. Model tests

### 2.1. Model Test Setup and Measurement System

This setup is suitable for simulating three-dimensional embankment model, which is helpful to deepen the understanding of load-transfer mechanism and deformation characteristics of pile-supported reinforced embankment. In addition, the model test setup adopts a separate structure, and the horizontal reinforcement is fixed at the connection between the upper box and the lower box, which is conducive to enhancing the lateral restraint of the reinforcement and exerting the membrane effect of the reinforcement.

The model test setup consists of an upper box, a lower box, and four model piles. The details of this setup are shown in Figure 2. The size of the setup is 1000 mm × 1000 mm × 1700 mm (length × width × height). The model pile is 450 mm tall overall. The pile cap is 250 mm in length by 250 mm in breadth.

The measurement System consists of single-point displacement meters and soil pressure cells. The layout of the components is also shown in Figure 2. d1 and d2 are single-point displacement meters used to measure the pile-subsoil relative displacement. d1 and d2 are situated at the mid-span of the subsoil between four piles and two piles, respectively. P1–14 are soil pressure cells, which are diaphragm strain-gauged type. P1 and P2 are, respectively, located at the pile cap's center and the corner. P3 is situated in the center of the subsoil surface between two piles. Between four piles, in the middle of the subsoil surface, is P4. P5–14 is distributed along the central axis of the embankment. All the soil pressure cells have been calibrated before use.

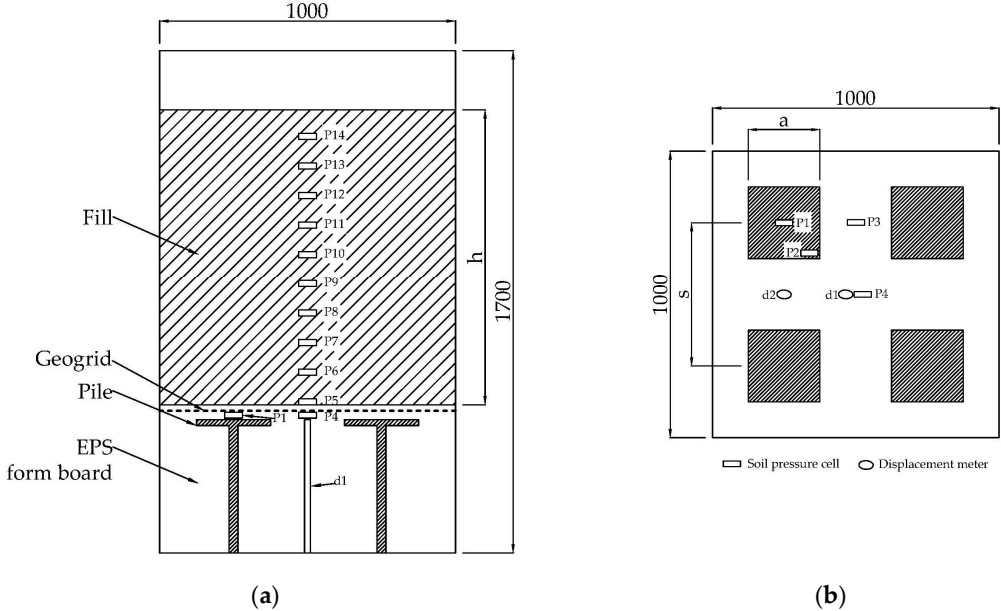

    (**a**)                                      (**b**)

**Figure 2.** Layouts of test setup (unit: mm), (**a**) Side view; (**b**) Top view. (h is the embankment height; a is the width of the pile cap; s is the pile spacing.).

### 2.2. Materials for Model Tests

The materials required for the model test are as follows: medium-fine sand, filaments, EPS foam board, and nonwoven geotextile. Medium fine sand is used as the embankment filling; nonwoven geotextile is used as the reinforcement, and EPS foam board is used to simulate a soft soil foundation.

It is known from the gradation tests that the coefficient of uniformity $C_u$ of the medium-fine sand is 1.75, the coefficient of curvature $C_c$ is 0.88, and the grading curve is shown in Figure 3. In addition, the particle proportion $G_S$ of the medium-fine sand is 2.643, and it's maximum dry density is 1.79 g/cm$^3$. The EPS foam board has a compressive strength of 200 $kPa$. The ultimate tensile strength of nonwoven geotextile is 30 kN/m. The above date is obtained according to JTG 3403-2020 Test Methods of Soils for Highway Engineering and Geotextiles [24] and GB/T 15788-2017 Geosynthetics-Wide-width tensile test [25].

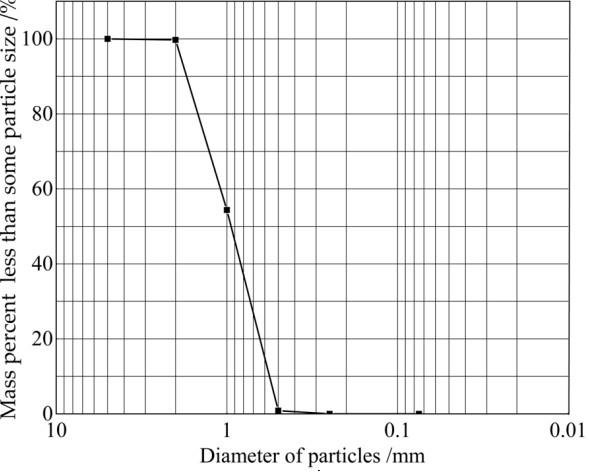

**Figure 3.** Grading curve of the sand.

The friction angle of medium-fine sand is measured to be 34.4° by direct shear tests. Direct shear tests are also performed on medium-fine sand mixed with 35 mm long filaments. The results suggest that when the weight percentages of the filaments are 0.1% and 0.22%, respectively, the filling cohesion is 11 kPa and 25 kPa. The cohesion of fill is changed by the internal restraint effect of filaments. However, If the filaments are too much, it is possible to form weak fiber structures in the soil, which hinder the exertion of soil friction strength, just like the low shear strength of soil with 0.22% filaments. Figure 4 shows the shear strength envelope.

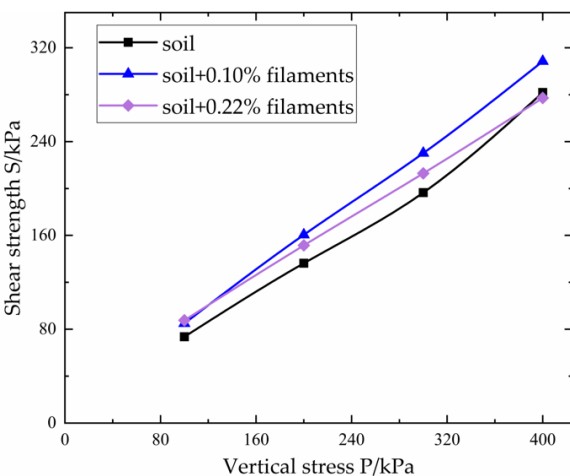

**Figure 4.** Shear strength curve.

### 2.3. Scheme of Model Test

Table 1 summarizes all of the model tests.

**Table 1.** Model test program.

|      | $c$ (kPa) | $s$ (mm) | $a$ (mm) | $h$ (mm) |
|------|-----------|----------|----------|----------|
| T1   | 0         | 500      | 250      | 1000     |
| T2   | 11        | 500      | 250      | 1000     |
| T3   | 25        | 500      | 250      | 1000     |

$c$: cohesion.

### 2.4. Steps of Model Test

1) Preparation of a series of instruments and equipment required for the model tests;
2) Placing model piles and EPS foam board;
3) Arranging the single-point displacement meter in the middle of the subsoil and attaching it to the comprehensive tester;
4) Connecting the soil pressure cells with the dynamic strain acquisition instrument and zeroed by software;
5) Burying soil pressure cells and laying reinforced cushion;
6) Filling the embankment into ten layers and burying the soil pressure cells after each layer is filled. In addition, the compaction degree is controlled by the quality of each filling layer.

Figure 5 is a fragment of the experiment.

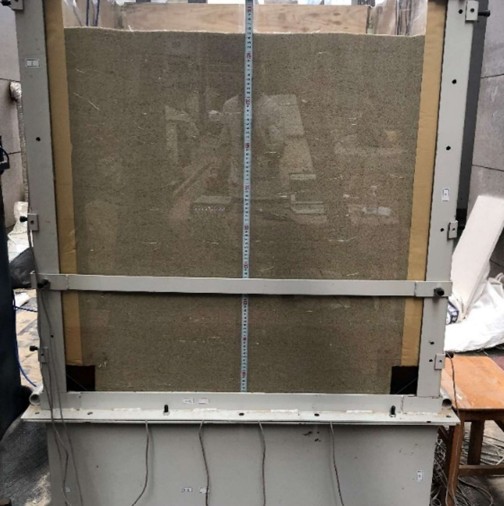

**Figure 5.** The filling of model tests.

## 3. Model Test Results

### 3.1. Stress State at the Bottom of the Embankment

Figure 6 depicts the changing process of soil pressure at the embankment's bottom during tests. Curves of soil pressure above pile caps and subsoil are shown in Figure 6a,b, respectively.

We can know from Figure 6a: (1) The soil pressure above the pile cap is not evenly distributed, and it is higher at the corner of the cap than it is in the middle. (2) As filling cohesiveness increases, the differential in soil pressure between the pile caps' corner and center reduces. (3) The soil pressure above the pile caps always maintains an enormous growth rate.

It is evident from Figure 6b that: (1) Compared to the soil pressure above the pile cap, the soil pressure above the subsoil is substantially lower. (2) The soil pressure is a little higher in the center of the subsoil between two piles than it is between four piles. (3) The soil pressure above the subsoil increases rapidly before the embankment height reaches 0.6 m and then gradually tends to be gentle.

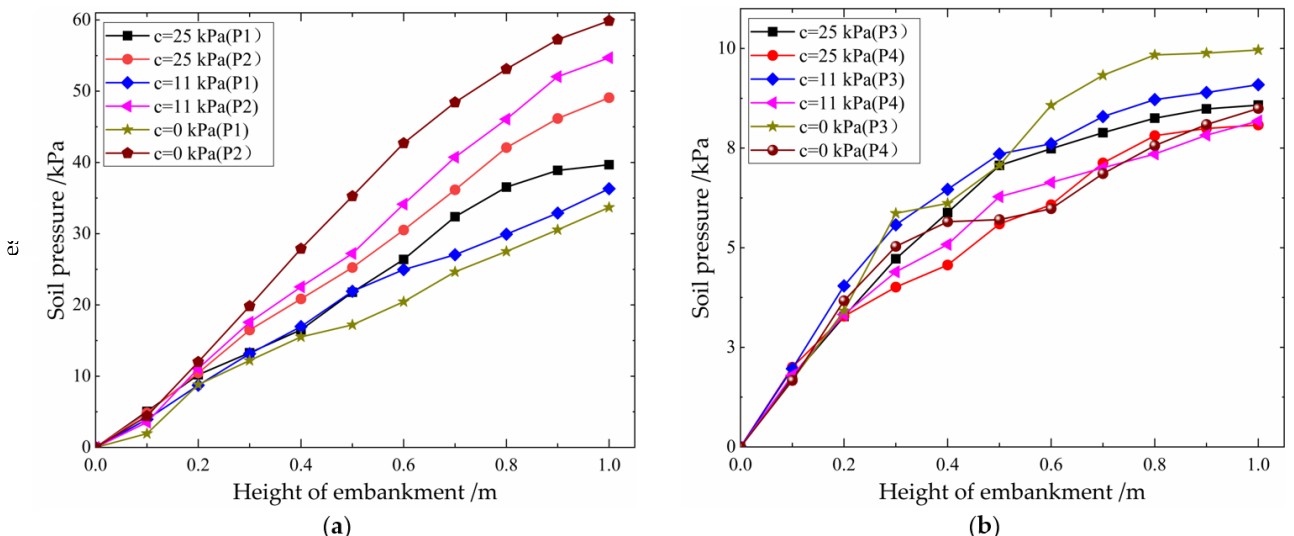

**Figure 6.** Variation in soil pressure at the embankment's bottom, (**a**) soil pressure above pile cap; (**b**) soil pressure above subsoil.

The redistribution of stress is a sign of soil arching, and soil arching is essential for enlarging the embankment load shared by the piles. The load distribution between the pile caps and the subsoil eventually differs significantly. The distinction is caused by: At first, the difference in stiffness between the piles and the subsoil is what causes the pile-subsoil settlement to differ; the shear stress in the fill is induced to hinder the relative movement in embankment filling. As a result, the pile caps are bearing the majority of the embankment's weight. The load distribution at the bottom of the embankment changes dramatically as a result of the modified load-transfer mechanism in the embankment. At the same time, the cohesion change will affect the value of shear stress on the slip surface as a non-negligible part of the shear stress. The load transfer and distribution are both impacted by the change in primary stress direction that occurs when shear stress changes. Lastly, a presentation of the variation in soil pressure above the pile cap is made.

### 3.2. Distribution of Vertical Stress

Figure 7 depicts the vertical stress curve within the embankment for various filling cohesions. That is apparent.

1) In all tests, the curve of vertical stress of the embankment is deflected;
2) In some situation when the distance is less than 0.3 m from the bottom of the embankment, the value of vertical stress increases with the increase in cohesiveness;
3) The deflection height on the curve of vertical stress is less affected by the cohesion, and the deflection height is about 0.3 m.

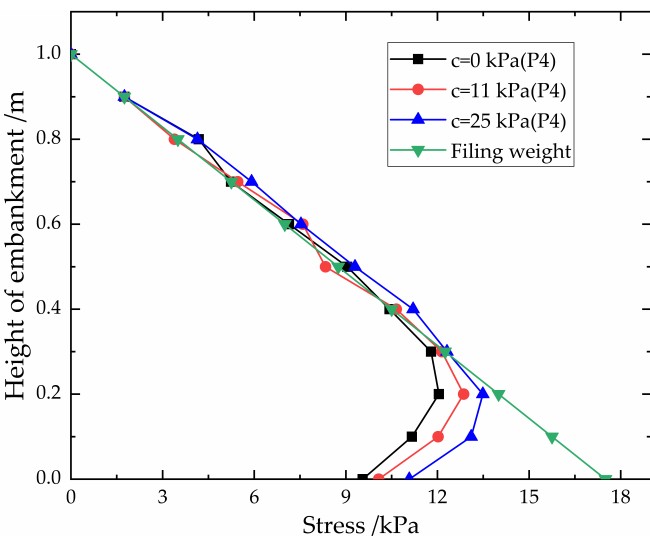

**Figure 7.** Effect of filling cohesion on Vertical stress in embankments.

The load of the embankment is transmitted by the contacts and force between the particles. The status (e.g., magnitude and orientation) of contacts and force between the particles have been changed due to the relative motion between particles, which results in the deflection of the vertical stress. Therefore, it is of great help to clarify the distributional state of stress in embankments for studying the load-transfer mechanism in embankments. Figure 7 shows that the interaction between particles rises with the increase in cohesion, as well as the ability of the embankment to resist deformation, but not the trend of relative movement in embankment filling and the deflecting angle of the contact force. Within the range of 0.3 m above the embankment's bottom, the vertical stress rises in proportion to the increase in filling cohesion. The deflection height of vertical stress, however, is little affected by the cohesion of the embankment filling.

### 3.3. Pile-Subsoil Relative Displacement

In essence, the pile-subsoil differential settlement is what causes the soil arching in embankments. Figure 8 displays the monitoring findings of the pile-subsoil differential settlement in model experiments. Figure 8 demonstrates:

1) The pile-subsoil differential settlement continues to grow during the embankment filling process, and the rate of growth gradually decreases. The final values of pile-subsoil differential settlement at the mid-span between four piles are 4.89 mm, 4.69 mm, and 4.58 mm, respectively, and the final values of pile-subsoil differential settlement at the mid-span between two piles are 3.85 mm, 3.69 mm, and 3.55 mm respectively. The different settlement at the mid-span between four piles is more extensive than between two piles. The pile spacings are always 500 mm. However, there is a difference between the spacing of diagonal adjacent piles and horizontal adjacent piles. As a result, the subsoil between four piles needs to bear a greater load.

2) As filling cohesiveness increases, the pile-subsoil differential settlement reduces. The different settlement at the mid-span between four piles in Test 2 and Test 3 decreased by 4.09% and 6.34%, respectively, and the different settlement at the mid-span between two piles in T2 and T3 decreased by 4.11% and 7.79%, respectively, compared with T1. It confirms that the pile-subsoil differential settlement, which is also caused by the variable spacings of diagonal adjacent piles and horizontal adjacent piles, is more susceptible to cohesiveness change at the mid-span between four piles than it is at the mid-span between two piles.

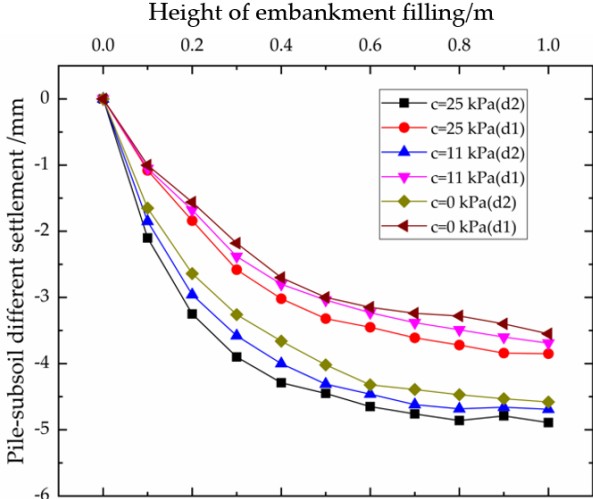

**Figure 8.** Effect of filling cohesion on pile-subsoil differential settlement.

### 3.4. Pile-Subsoil Stress Ratio

The average soil pressure at the top of the pile caps divided by the average soil pressure above the subsoil is known as the pile-subsoil stress ratio, which represents the features of pile-subsoil load efficiency. It is also a crucial indicator for embankment-settlement calculation, foundation-bearing capacity, and stability analysis. Figure 9 depicts the curve for the pile-subsoil stress ration.

(1) Before the relative height of the embankment ($h/(s - a)$) reached 2.4, the pile-subsoil stress ration grew quickly, and after that it tended to remain constant.

(2) As cohesiveness rises, the pile-subsoil stress ratio decreases under the embankment of the identical height.

The effect of cohesion on the stress distribution in the embankment is what causes the pile-subsoil stress ratio decreases as filling cohesion increases. Furthermore, the settlement and deformation of the embankment are also affected by the filling cohesion,

which changes the load distribution at the embankment's bottom. The slip surfaces are formed, in the embankment filling, because of the pile-subsoil different settlement. The of the active soil's self-weight load is transmitted to the adjacent and transverse soil, and the pile-soil stress ratio is greater than 1, because of the friction and occlusion between the soil particles. The increase in filling cohesiveness increases the particle contact force and the embankment's integrality. Therefore, pile-subsoil different settlements and the relative movement between the particles are weakened, and less load is transmitted to the pile caps. The pile-subsoil stress ratio reduces as a result.

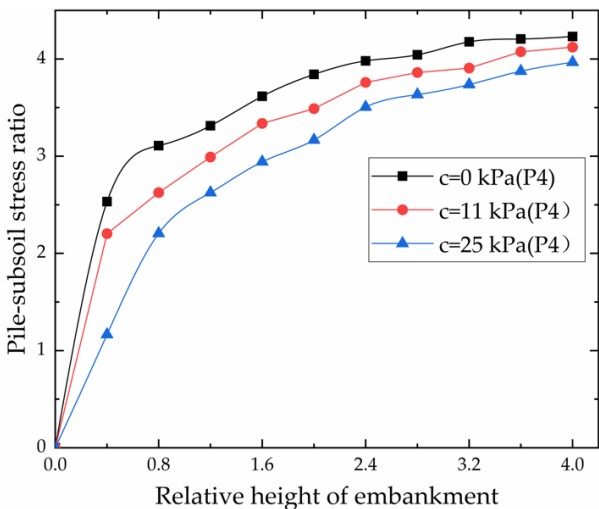

**Figure 9.** Effect of filling cohesion on pile-subsoil stress ratio, relative height of the embankment is $h/(s-a)$

## 4. FEM Numerical Analysis

### 4.1. FEM Calibration and Verification

The numerical model is created by ABAQUS, finite element method (FEM), based on the model test, as shown in Figure 10. The linear elastic model is used for piles, subsoil, and reinforcement. The Mohr-Coulomb model is used for the embankment and cushion. The regular contact surfaces between the piles and subsoil adopt hard contact, and the tangential connection adopts penalty. The interaction between reinforcement and filling is embedded. Horizontal constraints are applied around the model, and vertical constraints are applied at the bottom of the model [26]. The material parameters are shown in Table 2, and the test program is shown in Table 3.

**Table 2.** Material parameters.

| Parameter | Fill | Geogrid | Subsoil | Pile |
|---|---|---|---|---|
| E (MPa) | 15 | 0.5 | 0.2 | 200,000 |
| Internal Friction Angle (°) | 34.4 | - | - | - |
| Volume Weight (kN/m³) | 17.5 | - | 0.15 | 25 |
| Pisson Ration | 0.35 | 0.2 | 0.3 | 0.2 |

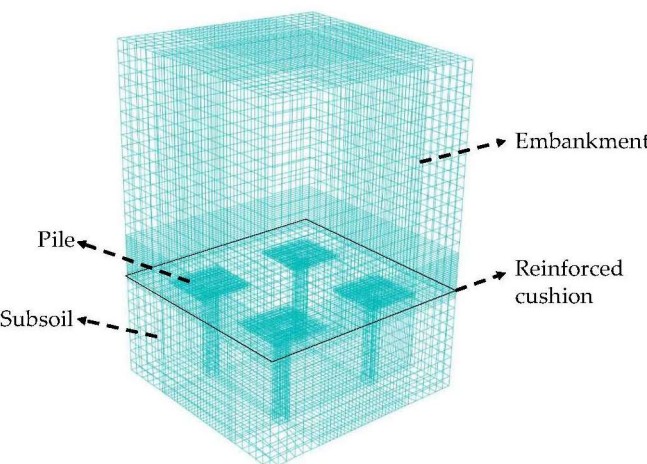

**Figure 10.** Three-dimensional finite element model of the pile-supported reinforced embankment.

**Table 3.** Test program.

| NO. | $c$ (kPa) | $s$ (mm) | $a$ (mm) | $h$ (mm) |
|-----|-----------|----------|----------|----------|
| M1 | 5 | 500 | 250 | 1000 |
| M2 | 15 | 500 | 250 | 1000 |
| M3 | 25 | 500 | 250 | 1000 |
| M4 | 45 | 500 | 250 | 1000 |

The curves of the vertical stress obtained, which are distributed along the central axis of the embankment, by the FEM test M3 and the model test T3 are shown in Figure 11. There is some discrepancy between the two tests. The reasons are:

1.  The stiffness of soil pressure cells used in the model tests is more prominent than it of the soil, and its diameter is 78 mm. This results in a concentration of stress above the soil pressure cells. Due to the aforementioned factors, the measured results of soil pressure cells are too large.
2.  It can be seen from Figure 11 that the measurement discrepancy is slight when the distance from the embankment bottom exceeds 0.4 m, but not when the distance from the embankment bottom is less than 0.4 m. The soil pressure cells are shifted during the process of embankment filling, because of the relative movements between the embankment filling. Soil pressure cells' measurement accuracy is hampered.
3.  The compressive properties of the medium-fine sand lead to a continuous decrease in the compressibility of the filling during compression. At the same time, the material's elastic modulus remains unchanged during the process of numerical simulation, which leads to the difference between the two tests.

Although there are differences between the finite element method and model tests, both can reflect the internal stress redistribution of embankment caused by pile-subsoil differential settlement. And the height of vertical stress deflection in both tests is about 0.4 m. Therefore, the finite element method still has specific guiding significance for analyzing soil arching.

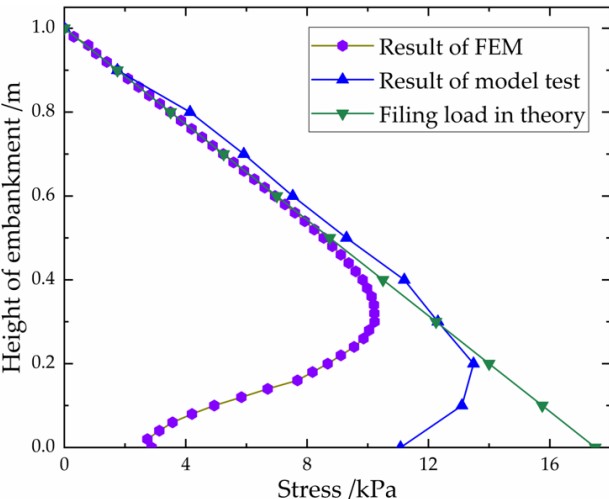

**Figure 11.** Verification of the finite element method.

### 4.2. Soil-Arching Geometry

The inflection point of the vertical stress curve determines the soil-arching axis. Figure 12 describes the soil-arching axis with the influence of embankment filling cohesion. In Figure 12, the ordinate is the height of the inflection point of the vertical stress curve, and the abscissa represents the separation from the pile cap's center (the star symbol in Figure 12). The soil-arching curve variation in Figure 12 exhibits the following traits:

(1) The height of the soil-arching axis between the diagonal adjacent piles is higher than between the horizontal adjacent piles. In tests M1, M2, M3, and M4, the height of the soil-arching crown between the diagonal adjacent piles increases by 38.5%, 32%, 30%, and 28.5% compared with that between the horizontal adjacent piles. It can be seen that there are significant differences in soil-arching morphology by comparing Figure 12a,b. That is because the net spacing $(s - a)$ between diagonal adjacent piles is larger than it is between the horizontal adjacent piles, as well as the overburden load of subsoil, which leads to sizeable pile-subsoil differential settlement, and thus the height of the soil-arching axis increases.

(2) As cohesiveness rise, the height of the soil-arching crown falls. Compared with the test M1, the height of vault between horizontal adjacent piles in tests M2, M3, and M4 decreases by 3.85%, 7.69%, and 9.62%, and the height of vault between diagonal adjacent piles decreases by 8.33%, 13.89%, and 16.11%. These findings suggest that as filling cohesion increases, the effect of cohesion on the soil-arching form is gradually weakened. When the filling cohesiveness changes, the soil-arching height between the diagonal adjacent piles is more sensitive than it is between the horizontal adjacent piles.

(3) The height of the soil-arching crown in Figure 12a is about $0.94(s - a) \sim 1.04(s - a)$, as that is about $0.87\sqrt{2}(s - a) \sim 1.0\sqrt{2}(s - a)$ in Figure 12b. All in all, the height of the soil-arching crown is roughly 1.0 times the net spacing in the embankment.

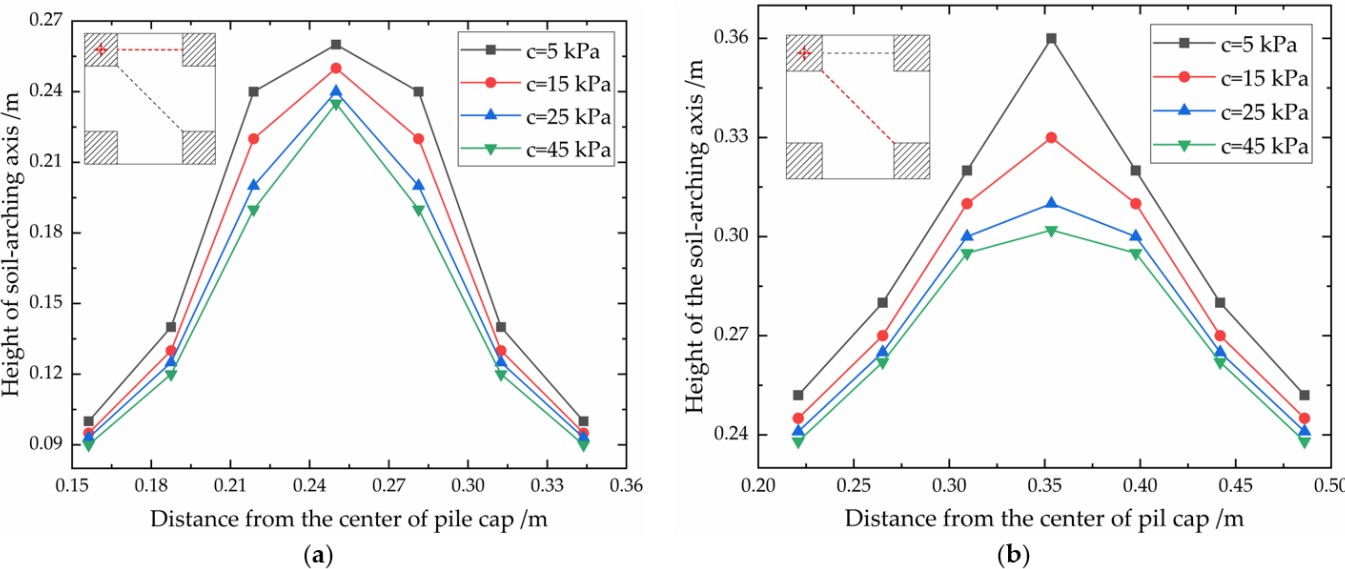

**Figure 12.** The soil-arching axis in the embankment, (**a**) the soil-arching crown between the horizontal adjacent piles; (**b**) the soil-arching crown between the diagonal adjacent piles.

### 4.3. Deformation Behavior

The internal displacement and stress distribution of embankment obtained from each test have a similar law. The M3 test is used as an example in the next section to examine the embankment's deformation patterns. Figure 13 reveals the distribution of vertical displacement at various filling heights. The embankment is divided into different areas according to the distribution of vertical displacement, as shown in Figure 13. From Figure 13, it is clear that the embankment's internal displacement exhibits the traits listed below:

(1) When the embankment's height is relatively low, sliding surfaces penetrate the entire embankment, causing localized differential settlement at the embankment surface, which is a significant undesirable settlement for the building of roads and railroads. The area with a substantial settlement inside the embankment is outlined by a triangle, and the areas surrounding the triangle are two non-overlapping inverted triangles.

(2) Two inverted triangular areas begin to overlap with the increase of embankment-filling height. The different settlement on the embankment surface is lessened when the soil-arching effect develops gradually.

(3) With the increase in filling height, planes with the same settlement appear in the embankment. The embankment height is now adequate for the development of the whole soil arching. It also demonstrates that the soil-arching effect can inhibit the uneven settlement of the embankment surface.

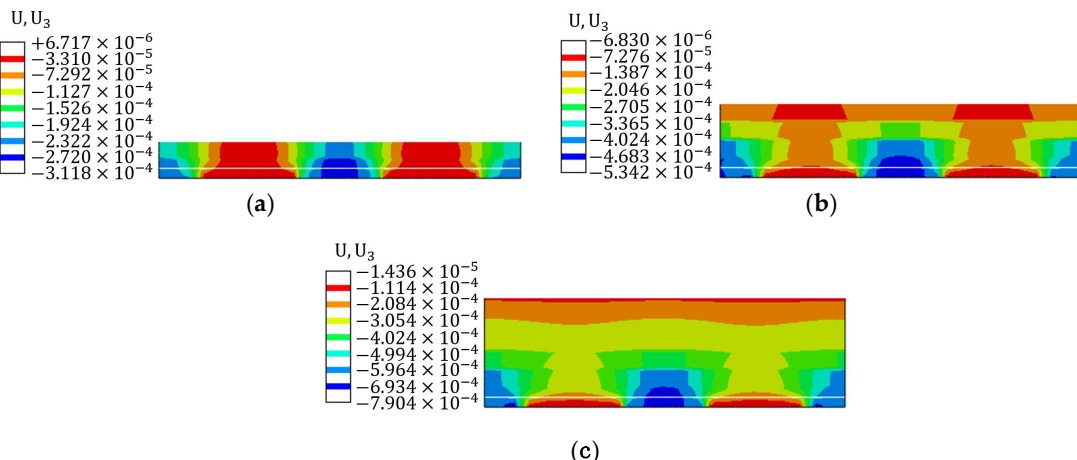

**Figure 13.** Vertical displacement distribution inside of embankment, (**a**) h = 50 mm; (**b**) h = 150 mm; (**c**) h = 250 mm.

The embankment is divided into three zones, which respectively represent the central sliding zone ($a_1$), secondary sliding zone ($a_2$), and stable zone ($a_3$) under the condition of small *h*, as shown in Figure 14a. Relatively large vertical displacements occur in the zone of $a_1$ and $a_2$, and the soil in the $a_1$ starts to slide earlier than that in the $a_2$. The lateral constraint of the secondary sliding area is weakened due to the movement of the particles in the $a_1$. And the sliding friction at the contact surfaces between $a_1$ and $a_2$ is overcome by the particles in $a_2$. Then the particles begin to move. Due to the occlusion and embedding action between particles, the differential settlement in the embankment gradually weakens when filling height is increased.

The soil arching has fully formed when the embankment height is significant, as seen in Figure 14b. The embankment is separated into five zones based on vertical settlement: the sliding zone ($b_1$), compaction zone ($b_2$), stable zone ($b_3$), transition zone ($b_4$), and equal settlement zone ($b_5$). The downward sliding tendency occurs in the zone of $b_1$ and $b_2$ due to the overlying load. Meanwhile, the differential settlement in embankments is caused, which decreases with the increase of the distance from the embankment bottom, by the vertical constraint of the pile caps on the embankment. Comparing Figure 14a,b, It is evident that when the embankment height is low, the soil-arching efficiency cannot be thoroughly aroused and the differential settlement diffuses to the surface of the embankment. However, when the filling height reaches a particular level, the entire soil arching will occur in the embankment. There is a non-different settlement on the surface of the embankment due to the effective displacement control controllability of the soil arching. This further demonstrates that the soil-arching effect is crucial in regulating the uneven settlement of embankments.

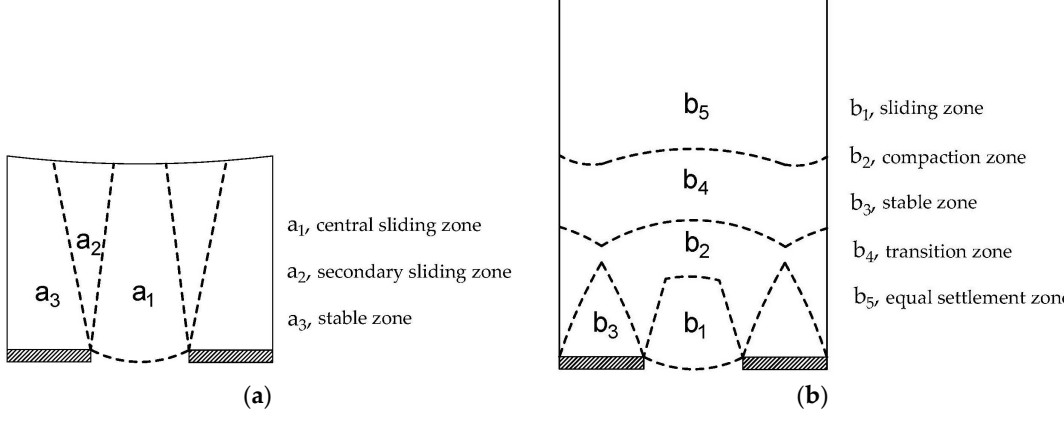

**Figure 14.** Schematic diagram of area division in the embankment. (**a**) Low embankment height, (**b**) High embankment height.

### 4.4. Equal Settlement Plane

The vertical displacement on the multiple cross sections in the embankment is, in the test of M1, M2, and M4, extracted for analysis, as shown in Figure 15. The planar diagram of the extracted path of the data is also shown in Figure 15. In addition, the data from M2 are omitted because of the similar deformation rules with M1, M3, and M4. The characteristics of vertical displacement can be seen in Figure 15.

(1) The variability of the filling cohesiveness has not essentially altered the deformation pattern of pile-supported reinforced embankment. Meanwhile, note that the above conclusion is related to the fact that the embankment is composed of granular material. Although the cohesion of the embankment filling has been increased, the granular material's properties have not been fundamentally changed. The equilibrium condition between the soil particles at the bottom of the embankment will be broken because of the subsoil's settlement. The cooperative movement of nearby particles will result in the uneven settlement inside the embankment.

(2) The change in filling cohesion has little impact on the displacement of soil particles, and when the filling cohesion increase, there is a slight reduction in the differential settlement at the same height in embankments. For example, the maximum settlement values in the M1, M3, and M4 are 2.42 mm, 2.37 mm, and 2.35 mm, respectively, when the embankment height h = 0.5.

(3) The progressive formation of equal-settlement planes during the filling of embankments is only marginally impacted by changes in filling cohesion. The equal settlement surface heights monitored from the tests M1, M3, and M4 are 0.36–0.4 m, 0.30–0.33 m, and 0.30–0.33 m, respectively. In addition, combined with the soil-arching height mentioned in the previous section, the heights of the soil-arching crown from tests M1, M3 and M4 are 0.36 m, 0.31 m, and 0.302 m, respectively. Altogether, the soil arching and the equal-settlement plane are approximately identical in height.

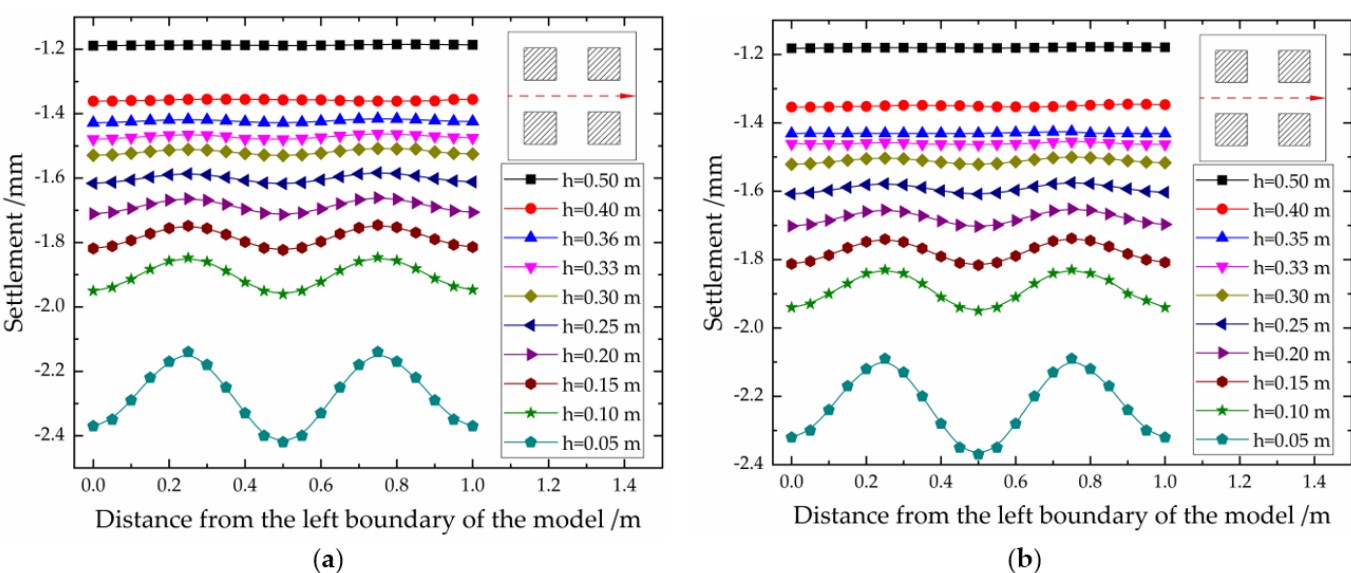

(**a**)　　　　　　　　　　　　　　　　(**b**)

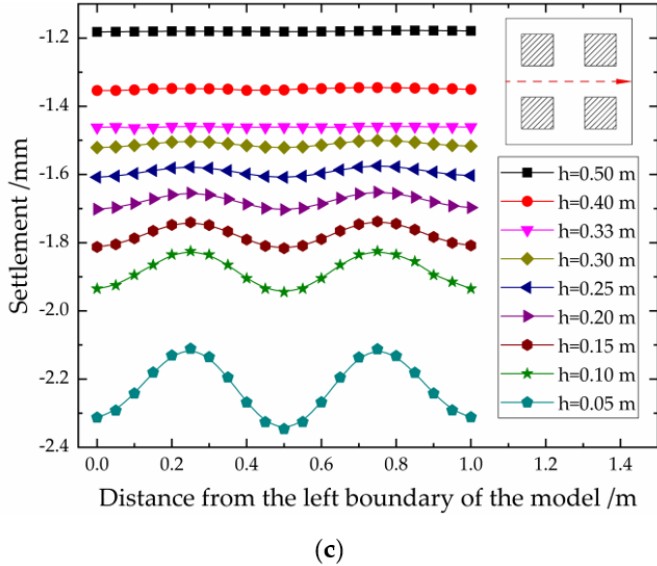

(**c**)

**Figure 15.** The curve of Vertical displacement in embankment, (**a**) result of M1; (**b**) result of M3; (**c**) result of M4.

## 5. Conclusions

Model tests and finite element method have been carried out to analyze the mechanism of load transfer, deformation behavior, and soil-arching geometry with different embankment-filling cohesion in this study. The controllability of the filling cohesion is achieved by filaments, which are incorporated into the sand with varying percentages of weight. Following conclusions can be drawn,

1.  When the filling is completed, changing the filling cohesion affects the load-transfer efficiency. It has more influence on the load distribution above the pile cap, while less on the load distribution above the subsoil. As the filling cohesion increases, the soil-pressure difference between the corner and center of the pile caps decreases.
2.  The filling cohesion affects the pile-subsoil load-transfer efficiency during the embankment filling process. The pile-subsoil stress ratio falls with the increase of filling cohesion. The curves of the pile-subsoil stress ratio grow rapidly at the beginning and then tend to stabilize at the later stage.
3.  With a rise in filling cohesion, the pile-subsoil differential settlement and soil-arching height both decrease. Pile-soil differential settlement at the mid-span of four piles is greater than at the mid-span of two piles. Meanwhile, the soil-arching height between the diagonal adjacent piles is higher than that between the horizontal adjacent piles. The above results the net pile spacing between diagonally adjacent piles is greater than the net spacing between horizontally adjacent piles.
4.  The soil-arching crown in the embankment is roughly the same height as the net pile spacing. The influence of filling cohesion on the soil-arching height is more significant only in the case of less cohesive working conditions.
5.  The vertical displacement inside the embankment decreases with the increase of filling cohesion. The equal settlement plane and the soil arching both have the same height.

In summary, the indoor model experiment and finite element method are used as the research tool in this paper. The influence of cohesion (the filling property) on mechanical behavior and deformation behavior of the pile-supported reinforced embankments is further investigated. The investigation is under the 3D condition to better understand the working properties of pile-supported reinforced embankment in the case of square pile placement. The research results show that the filler cohesion greatly influences the stress

redistribution in the embankment. And its influence on the internal deformation pattern of the embankment is small.

**Author Contributions:** Writing-original draft preparation, D.Z.; funding acquisition, G.Y; writing-review, X.W.; project administration, Z.W.; Methodology, H.W. All authors have read and agreed to the published version of the manuscript.

**Funding:** This research was supported by the National Key R&D Program of China (Grant No. 2022YFE0104600); the National Natural Science Foundation of China (Grant No. 52079078); the Key R&D Projects in Hebei Province (Grant No. 20375504D); and the Hebei Provincial Department of Transportation Science and Technology Project (Grant No. RW-202014).

**Institutional Review Board Statement:**  Not applicable.

**Informed Consent Statement:**  Not applicable.

**Data Availability Statement:** The data used to support the findings of this research are available from the corresponding author upon request.

**Conflicts of Interest:** The authors declare no conflict of interest.

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
