# Peer review of "Analysis of Load Transfer and the Law of Deformation within a Pile-Supported Reinforced Embankment"

_applsci, doi:10.3390/app122312404_

Round 1

Reviewer 1 Report

1. In the Abstract, What do you mean the functional properties? You may introduce the exact functional properties briefly here. Actually I did not find any functional properties in the manuscript.

2. Lots of experimental results are presented in the Abstract without explanation and mechanism discussion. The abstract should be greatly improved on this.

3. English expression should be improved in the section of Introduction. Several paragraph can be integrated.

4. You may simply introduce the superiorities of the presented model in comparison to previous studies.

5. In figure 3, could you please explain the low shear strength of soil with 0.22% filaments.

6. Correct the Kpa to kPa in all the figures.

Reviewer 2 Report

In this paper, an Analysis of load transfer and the law of deformation within a

pile-supported reinforced embankment was carried out. The paper is of interest. But some questions and weak points should be mentioned. In most cases, some parts need a better explanation. A lot of needed information is missed, mostly in the methodological part. Everything should be clear for readers.

1- MATERIAL AND METHODS: A flowchart of the research methodology and its explanations is needed for this study.

2-Based on the research similarity check(attached) for this study is 27% and should e less than 18%.

3-MATERIALS AND METHODS: The tests were performed according to which standard or procedure? You need to add this information to the manuscript.

4- Abstract. Please, include exact results and conclusions with obtained values.

5- In fig10:verification of FEM model: The horizontal vector( stress/kPa), why is it different between (the FEM model test and FEM and Filling test theory)?

6- In fig 4 and 9:The dimensions of the model with details should be illustrated in it.

7- in Table 2, some property amount is missing.

8- English editing is needed.

Round 2

Reviewer 1 Report

The manuscript can be accepted after proper modification.

Reviewer 2 Report

Accept in present form